# A Reliable Prognosis Approach for Degradation Evaluation of Rolling Bearing Using MCLSTM

**DOI:** 10.3390/s20071864

**Published:** 2020-03-27

**Authors:** Gangjin Huang, Hongkun Li, Jiayu Ou, Yuanliang Zhang, Mingliang Zhang

**Affiliations:** School of Mechanical Engineering, Dalian University of Technology, Dalian 116024, China; huanggj@mail.dlut.edu.cn (G.H.); oujy@mail.dlut.edu.cn (J.O.); zylgzh@dlut.edu.cn (Y.Z.); zml1314mln@mail.dlut.edu.cn (M.Z.)

**Keywords:** Gaussian process latency variable model, multiple convolutional long short-term memory network, rolling bearing, remaining useful life

## Abstract

Prognostics and health management technology (PHM), a measure to ensure the reliability and safety of the operation of industrial machinery, has attracted attention and application adequately. However, how to use the monitored information to evaluate the degradation of rolling bearings is a significant issue for its predictive maintenance and autonomic logistics. This work presents a reliable health prognosis approach to estimate the health indicator (HI) and remaining useful life (RUL) of rolling bearings. Firstly, to accurately capture the degradation process, a novel health index (HI) is constructed based on correlation kurtosis for different iteration periods and a Gaussian process latency variable model (GPLVM). Then, a multiple convolutional long short-term memory (MCLSTM) network is proposed to predict HI values and RUL values. Finally, we perform experimental datasets of rolling bearings, demonstrating that the presented method surpasses other state-of-the-art prognosis approaches. The results also confirm the feasibility of the presented method in industrial machinery.

## 1. Introduction

Since the development of manufacturing technology, the structure of machinery tends to be integrated, complicated and intelligent; meanwhile, health management technology of mechanical equipment has received increasing research attention [1,2,3]. The core problem of health management is how to predict the remaining useful life (RUL) through the monitoring data accurately, and then determine the optimal maintenance opportunity. By minimizing the economic cost or risk of equipment failure, condition-based predictive maintenance and autonomous maintenance can be realized [4]. Therefore, it is still a great challenge for RUL prediction research of rolling bearings based on vibration data [5,6].

Nowadays, the RUL prediction methods of rolling bearings can be divided into two categories, i.e., model-based and data-driven approaches. Model-based approaches are mainly based on fracture mechanics, damage mechanics, stress, strain and other methods [7,8,9]. El-Tawil et al. [10] developed a prognostic methodology based on nonlinear damage to predict the RUL of the system. Pu et al. [11] studied the effect of sliding motion on the contact fatigue life of bearings. Wang et al. [12] investigated a hybrid prognosis approach for machine condition prognosis of bearings in a wind turbine. Cubillo et al. [13] presented a potential physics based on the prediction model for rotating machinery to describe the degradation process of the machinery. Chadli et al. [14] developed a novel method based on the design of distributed state estimation and fault detection and isolation filters, and its effectiveness was illustrated by a numerical example. Qian et al. [15] presented a modified Paris crack growth model to describethe bearings’ defect propagation on a slow-time scale, and phase space warping was enhanced by a multi-dimensional auto-regression model for RUL prediction. Chadli et al. [16] studied a polynomial fuzzy filter to solve the issues of fault detection. The model-based approaches can get deep into the essence of the target machine and get more accurate prediction results when predicting the degradation trend and remaining useful life. However, in the majority of engineering applications, it is difficult to establish a specific mathematical model that reflects the mechanical and physical laws of performance degradation.

The data-driven approaches can genuinely reflect the dynamic behavior of the monitoring object, and it is easy to adjust the model parameters and update the trend of performance degradation, which makes it develop and apply [17,18] rapidly. In general, data-driven RUL prediction includes four main processes [7]: data acquisition, feature extraction, degradation behavior learning and RUL estimation. An online performance evaluation method was proposed to predict the remaining life of a turbine [19]. Javed et al. [20] developed a novel approach using wavelet extreme-learning for RUL estimation task of bearings. Liu et al. [21] proposed an integrated framework to track the degradation state of bearings. With the development of sensor technology, data explosion has become a new problem of bearing remaining life prediction, which has made artificial intelligence technology develop rapidly in recent years. Zhang et al. [22] constructed a long short-term memory (LSTM) network to predict the RUL of mechanical equipment and verified the advantages of LSTM in RUL prediction by using C-MAPSS datasets. Zhao et al. [23] presented a recurrent neural network (RNN) means for RUL prediction based on trend characteristics, and the effect of the presented method outperformed the other latest methods. Wang et al. [24] proposed a novel recurrent convolutional neural network for RUL prediction of rolling element bearings and milling cutters. Chen et al. [25] employed a new gated recurrent unit, and the C-MAPSS datasets verified the robustness of the proposed method. Although these methods are widely used in machinery prognostics, they have their drawbacks. The deep learning model is still insufficient to extract the spatial and temporal characteristics of time series data. However, the recurrent neural network can effectively obtain the temporal characteristics, but ignores the spatial characteristics of time series data.

To solve the aforementioned problems, the current paper presents a novel estimation method for health indicator (HI) and RUL prediction of rolling bearings. Firstly, the correlation kurtosis of different iteration periods is extracted from the frequency-domain data, and then a novel HI is obtained by a Gaussian process latency variable model (GPLVM). Following that, a prognosis model, the so-called multiple convolutional long short-term memory (MCLSTM), is proposed to predict future HI and the RUL. Finally, the feasibility of the present approaches was validated by bearing experiment datasets.

The main contributions of this paper are summarized as follows:

(1) A special HI is constructed based on correlation kurtosis for different iteration periods and GPLVM. The HI can effectively depict the degradation process of rolling bearing. 

(2) A novel deep prognosis network, i.e., MCLSTM, is presented to mine the temporal–spatial correlation of rolling bearing, which improves the accuracy of the HI and RUL estimations considerably.

(3) We verify the capability of the proposed approach based on experimental datasets of rolling bearing. The results show that MCLSTM achieved better performance than the state-of-the-art prognosis approaches.

The remainder of the paper provides a theoretical background in Section 2. Section 3 presents the proposed deep learning model for prediction. The experimental results and thorough discussion are given in Section 4. Finally, Section 5 concludes the whole work.

## 2. Theoretical Background

### 2.1. HI Construct

In the field of fault diagnosis, kurtosis is an essential index to detect the health condition of mechanical equipment [26], which can be defined as follows:(1)$$Kurtosis=\frac{\frac{1}{n}\sum_{n=1}^{N}x_{n}^{4}}{{\left(\frac{1}{n}\sum_{n=1}^{N}x_{n}^{2}\right)}^{2}}$$
where x denotes the signal sequence, n denotes the sampling points. However, kurtosis is difficult to reflect the intensity of the specific periodic pulse signal in the vibration signal. To tackle this problem, correlation kurtosis (CK) [27] is proposed based on the kurtosis.
(2)$$CK_{M}(T)=\frac{\sum_{n=1}^{N}{\left(\prod_{m=0}^{M}x_{n-mT}\right)}^{2}}{(\sum_{n=1}^{N}x_{n}^{2}{)}^{M+1}}$$
where N denotes the number of sampling points of the signal, M denotes the number of the shift cycles, and T denotes the sensitive cycles. When T=0 and M=1, Equation (2) is equal to Equation (1). It can be found from Equation (2) that the CK has the characteristics of the periodic correlation function. 

Fourier transform the time-domain signal x(t) to frequency-domain signal y(t). Then calculate the CK value at different offset points, which can be formulated as:(3)$$y(t)=\int_{-\infty }^{+\infty }x(t)\exp(-i\omega t)dt$$
(4)$$FCKT_{M}(T)=\frac{\sum_{n=1}^{N}{\left(\prod_{m=0}^{M}y_{n-mT}\right)}^{2}}{{\left(\sum_{n=1}^{N}y_{n}^{2}\right)}^{M+1}}$$

As Equation (4) shows, T and M are unknown variables. If the number of offset points T is regarded as a variable, and M is defined as specific values, then a multi-dimensional matrix of CK values can be calculated by Equation (4).

### 2.2. Gaussian Process Latency Variable Model

Dimension disaster is a prevalent problem in statistics and machine learning [28]. Due to the high dimensionality of the data, a complex calculation is indispensable in the data processing. Therefore, it is vital to discover an excellent dimensionality reduction which can obtain the potential low dimensional structure information from high dimensional data [29]. GPLVM, a flexible Bayesian nonparametric model, has been applied to dimensionality reduction of the signal in the past few decades [30]. 

GPLVM assumes that the observed dataset $Y={\left[y_{1},y_{2},\cdots ,y_{N}\right]}^{T}\subset R^{D}$ is composed of n d-dimension data; the q-dimension dataset $X={\left[x_{1},x_{2},\cdots ,x_{N}\right]}^{T}\subset R^{Q}$ is obtained after dimension reduction. Then, the *i*th training sample is defined as follows [29]:(5)$$y_{i}=f(x_{i})+\varepsilon$$
where f denotes the nonlinear function with Gaussian process(GP) prior f~GP(0,K), ε denotes the noise with Gaussian distribution $\varepsilon \sim N(0,\delta^{2})$.

By using the Bayes theorem and integrating f, then we can know the marginal likelihood p(Y|X,θ), which is formulated as:(6)$$p(Y|X,\theta )=\prod_{j=1}^{D}\frac{1}{{\left(2\pi \right)}^{\frac{1}{2}}{\left|K\right|}^{\frac{1}{2}}}e^{-\frac{1}{2}y_{:,j}^{T}K^{-1}y_{:,j}}$$
where K denotes the kernel function, θ denotes the hyper-parameters of both kernel function and noise, and $y_{:,j}$ denotes the *j*th column of matrix Y. 

The object function of GPLVM can also be derived from probabilistic principal component analysis (PPCA). In PPCA, we assume that the parameter matrix W follows a prior Gaussian distribution [30], which is expressed as:(7)$$p(W)=\prod_{i=1}^{D}N(w_{i}|0,I)$$

After that, we can obtain the marginal likelihood probability by integrating out W, which can be described as follows:(8)$$p(Y|X,\beta )=\prod_{d=1}^{D}N(y_{:,d}|0,XX^{T}+\beta^{-1}I)$$

As Equation (8) shows, GPLVM can be equivalent to PPCA by replacing $XX^{T}$ in Equation (8) with a kernel matrix K [31].

### 2.3. LSTM Theory

As an improved recurrent neural network, LSTM neural network overcomes the drawbacks of an exploding gradient or the vanishing of RNN [32]. Additionally, it has a strong ability to capture the dynamic characteristic through cycles in sequential data. Moreover, LSTM adds a memory cell structure to the neurons in the hidden layer, which reduces the number of unknowns significantly.

In Figure 1, the LSTM hidden layer cell structure is presented, which can be made up of memory cell, input gate, output gate and forget gate. Among them, the memory cell remembers the activation value over any time intervals, and three gates regulate the input and output of information flow to the unit [33]. The update functions are formulated as follows:(9)$$\{\begin{array}{c} i_{t}=\sigma (W_{hi}h_{t-1}+W_{xi}x_{t}+W_{ci}⊙c_{t-1}+b_{i})\\ f_{t}=\sigma (W_{hf}h_{t-1}+W_{xf}x_{t}+W_{cf}⊙c_{t-1}+b_{f})\\ {\tilde{c}}_{t}=\tanh(W_{hc}h_{t-1}+W_{xc}x_{t}+b_{c})\\ c_{t}=f_{t}⊙c_{t-1}+i_{t}⊙{\tilde{c}}_{t}\\ o_{t}=\sigma (W_{ho}h_{t-1}+W_{xo}x_{t}+W_{\mathrm{co}}⊙c_{t}+b_{o})\\ h_{t}=o_{t}⊙\tanh(c_{t}) \end{array}$$
where ⊙ denotes the Hadamard product; σ denotes the sigmoid function; $i_{t}$, $f_{t}$, $c_{t}$ and $o_{t}$ denote the outputs of input gate, forgetting gate, memory cell and output gate, respectively; $W_{xi}$, $W_{hi}$ and $W_{ci}$ denote the weight matrix of input information, last time output and memory cell to input gate, respectively; $W_{xf}$, $W_{hf}$ and $W_{cf}$ denote the weight matrix of input information, last time output and memory cell to forgetting gate, respectively; $W_{xo}$, $W_{ho}$ and $W_{co}$ denote the weight matrix of input information, last time output and memory cell to output gate, respectively; $b_{i}$, $b_{f}$, $b_{c}$ and $b_{o}$ denote the bias of input information, forgetting information, memory cell and last time output, respectively and tanh(⋅) denotes the hyperbolic tangent function.

### 2.4. Architecture of the Proposed Network

Inspired by the receptive field of mammalian visual cortex cells, Convolutional neural networks (CNN) is a feedforward neural network in deep learning models. The convolutional layer and pooling layer are two critical layers of convolution neural network. In most cases, a convolutional layer is a useful tool for extracting sophisticated high-dimensional input features [34], and the pooling layer can reduce the space size of the input convolution layer and avoid overfitting of the network. CNN has a strong ability to extract features from the input, and it is widely applied in the field of image processing and text recognition [35,36].

As a deep learning model unit, convolutional long short-term memory (ConvLSTM) combined CNN and LSTM is specially designed for spatiotemporal sequence [37] and the structure is shown in Figure 2. LSTM has shown considerable power in dealing with temporal correlation issues, but it contains too much spatial redundancy. To overcome the limits, ConvLSTM integrates convolutional operation in the input-to-state and state-to-state transitions. Furthermore, ConvLSTM offers an outstanding generalization by increasing computational power, and the update functions are expressed as [38]: (10)$$\{\begin{array}{c} i_{t}=\sigma (W_{hi}*h_{t-1}+W_{xi}*x_{t}+W_{ci}⊙c_{t-1}+b_{i})\\ \begin{array}{l} f_{t}=\sigma (W_{hf}*h_{t-1}+W_{xf}*x_{t}+W_{cf}⊙c_{t-1}+b_{f})\\ {\tilde{c}}_{t}=\tanh(W_{hc}*h_{t-1}+W_{xc}*x_{t}+b_{c}) \end{array}\\ c_{t}=f_{t}⊙c_{t-1}+i_{t}⊙{\tilde{c}}_{t}\\ o_{t}=\sigma (W_{ho}*h_{t-1}+W_{xo}*x_{t}+W_{\mathrm{co}}⊙c_{t}+b_{0})\\ h_{t}=o_{t}⊙\tanh(c_{t}) \end{array}$$

As Equation (10) shows, all notations in Equation (10) are the same as those in Equation (9). The difference between them is that the $i_{t}$, $f_{t}$, and $c_{t}$ are all calculated by convolutional operation, i.e., ∗.

Due to the time–space correlation of the vibration signal in the degradation process of rolling bearing, a new prognosis model based on MCLSTM is proposed. Specifically, the MCLSTM is constructed by stacking multiple ConvLSTM units, and then batch normalization [39] technology is added between each ConvLSTM layer. After that, a dense layer is employed. Finally, the regression layer is built on the last layer.

## 3. The Proposed Framework

To evaluate the rolling bearings degradation, a novel prognosis approach is presented in this paper. Firstly, HI is carefully designed by correlation kurtosis for different iteration periods and GPLVM. Then, a novel prognosis network, i.e., MCLSTM is constructed by stacking multiple hidden layers. At last, experimental results confirm the feasibility and superiority of the presented method. The detailed steps of the proposed approaches are illustrated and the corresponding flowchart is shown in Figure 3.

(1) *Data acquisition*: The accelerometer is placed in the horizontal and vertical directions of the test bearing, and obtained the whole life experimental data of the rolling bearing. The sampling frequency of experimental data is 20 kHz, the sampling interval is 10 min, and the rotation speed is 2000rpm.

(2) *HI designed:* The correlation kurtosis values of frequency domain signals with different iteration periods by Equation (4) are calculated from the experimental data. GPLVM is utilized for fusing the correlation kurtosis of the different iteration periods, and the low dimensional sensitive features are selected as the HI of rolling bearing.

(3) *Proposed neural network:* The prognostics neural network is constructed by stacking multiple ConvLSTM units, and batch normalization technology is added between each ConvLSTM layer. The last two layers are the dense layer and regression layer. The network parameters are set as: the number of iterations is 200, the convolution kernel size is 3 × 3 the activation is sigmoid, the filter is 64, the loss function is mean square error, the optimizer is Adam.

(4) *HI and RUL prediction*: The designed HI is input into the proposed neural network to predict the future HI and RUL of rolling bearings. As a result, the health status of rolling bearings is evaluated, and the maintenance plan is provided with reference.

## 4. Case Verification

### 4.1. Data Description

In our case, a full-life-cycle experimental datasets of rolling bearings is indispensable to verify the feasibility of the proposed method. The experimental datasets were carried out on an accelerated aging platform by the center for Intelligent Maintenance Systems (IMS) [40], and the test rig structure is displayed in Figure 4. The test rig consists of four bearings of type Rexnord za-2115, mechanical spring system and motor. The mechanical spring system imposes radial load to the shaft and bearing, and the motor drives the test bearing to run. Two accelerometers were placed in the horizontal and vertical directions of each bearing to pick up vibration data. The sampling frequency of vibration data was 20 kHz, the sampling interval was 10 min, and the rotation speed was 2000rpm. Under the same experimental conditions, three groups of experiments were carried out, and each of which collected the vibration signal of four bearings from the normal to failure. In this paper, the experimental data of bearing 1 (outer ring failure) in the second group of experiments were selected for analysis, including 984 groups of vibration signals. 

### 4.2. Evaluation Indexes

For the sake of confirming the performance of the presented method, appropriate evaluation indexes are required. In this paper, we selected four indexes to judge the performance of the different models for remaining useful life estimation of rolling bearings, including the root mean square error (RMSE), the mean absolute percentage error (MAPE), mean absolute error (MAE) and the R-Square. The evaluation indexes are described as: (11)$$RMSE=\sqrt{\frac{1}{N}\sum_{i=1}^{N}(y_{i}-{\hat{y}}_{i}{)}^{2}}$$
(12)$$MAPE=\frac{1}{m}\sum_{i=1}^{m}\left|\frac{y_{i}-{\hat{y}}_{i}}{y_{i}}\right|$$
(13)$$MAE=\frac{1}{m}\sum_{i=1}^{m}\left|\left(y_{i}-{\hat{y}}_{i}\right)\right|$$
(14)$$R-Square=1-\frac{\sum_{i=1}^{m}{\left(y_{i}-{\hat{y}}_{i}\right)}^{2}}{\sum_{i=1}^{m}{\left(y_{i}-{\bar{y}}_{i}\right)}^{2}}$$
where $y_{i}$ and ${\hat{y}}_{i}$ denotes the actual value and predicted value. $\bar{y}$ denotes the mean value of the actual values. m denotes the total number of testing samples.

### 4.3. Health Stage Division Analysis

In this part, the health stage division is analyzed. To cope with the problem of RUL prediction, it is crucial to capture the degradation tendency from historical information. In this paper, we constructed a novel HI to describe the trend of bearing performance degradation. The time-domain waveform of the test rolling bearing is exhibited in Figure 5a. It can be observed that the amplitude of the vibration data starts to increase at about 7000 min. With the aggravation of the bearing fault, the amplitude of the vibration signal also increases until the experiment is stopped. As shown in Figure 5b, at the beginning of the bearing operation, the HI cure is stable without fluctuation. The HI value gradually rises, which can be observed at 5350 min, indicating incipient degradation of bearing. Then, there are prominent peaks and valleys in the curve. The peaks indicate that the degree of bearing wear has reached the maximum value at this stage. Subsequently, the curve begins to decline, which is the so-called healing phenomenon. The failure point of the rolling bearing gradually wears out and becomes smooth under the impact of the influence of impact force, which makes the HI curve decrease. According to the above analysis, we can determine the bearing damage evolution process through the HI curve and then estimate the current degradation stage of the bearing. Therefore, in this paper, we divide the stage of bearing performance degradation into four types: normal stage (stage 1), initial degradation (stage 2), moderate degradation (stage 3), and severe degradation (stage 4).

To further verify the inferences, we selected different stage vibration data for time-domain analysis and envelope demodulation analysis. Figure 6 shows the analysis results on the various stage data. Obviously, it can be observed in Figure 6 that there are obvious differences in the time domain waveform, and we can further analyze this through the envelope demodulation waveform. In the envelope demodulation waveform, we cannot find the characteristic fault amplitude at 550 min (Figure 6a), and the bearing is healthy. When the bearing runs to the 5350 min (Figure 6b), we can find the envelope amplitude appears 231.5 Hz, i.e., the characteristic frequency of outer ring fault and its second frequency, which indicates that the bearing has failed. With the operation time increase, the amplitude of fault characteristic frequency continues to increase, and the bearing degradation is further intensified. It should be noted that the amplitudes of 230.7 and 460.8 Hz in envelope demodulation waveform have nearly the same at 7040 and 8500 min, respectively (Figure 6c and Figure 6d). Simultaneously, we can also observe that the HI values of these two time points are approximately the same, which verifies the feasibility and necessity of HI.

### 4.4. RUL Prediction

In this section, some factors that may affect the prediction performance of MCLSTM are discussed in detail, including the iteration times and optimizer. Then, HI prediction value started from different time points are further discussed. At last, the performance of the proposed method for RUL estimation is analyzed.

(1) **Effect of the number of iterations**: The number of iterations is a critical hyper parameter in network structure, whose size may affect the computational cost of the proposed method. In this paper, we directly use multiple stacked units to form a prediction model, in which the filter is valued of 64, and the convolution kernel size is set to 3 × 3. The relationship between the loss function, i.e., prediction error and the number of iterations is shown in Figure 7. It can be observed that when the number of iterations is 200, the prediction error is already small enough. When the number of iterations is 200, the loss curve shows no fluctuation. In Figure 8, MAPE and operation time of different numbers of iterations are calculated, respectively. We can draw several conclusions: 1) As the number of iterations increases, the MAPE value decreases, indicating that the prediction performance is better. 2) As the number of iterations increases, the prediction model training time increases, but the time gap is only a few seconds. In total, it is proper that the number of iterations of the prediction model is 200.

(2) **Effect of the optimizer**: When the model updates the network weight and bias parameters, the choice of the optimizer directly affects the accuracy of the prediction results. It is critical to choose the optimizer of the prediction model when updating the model weights and bias parameters. We choose four typical optimizers (Adam, RmsProp, Adagrad and SGD) to train the prediction model. Furthermore, the values of MAE, RMSE, R-Square, MAPE and training time are calculated to verify the performance of the prognostic model, respectively. From Table 1, it can be seen that the Adam optimizer has better performance compared with the other optimizers, and the optimizer for model training is Adam in this paper.

(3) **HI and RUL prediction**: In this part, the HI and RUL prediction of rolling bearing is conducted. In Section 4.3, we divided the bearing full-life signals into four stages and used the proposed prediction model to predict the future HI. We input the previous samples at this time as training data into the model randomly, and the remaining samples are regarded as testing to detect the model capacity. It can be seen from Figure 9 that the prediction starts from three different moments when T = 5350 min, T = 7040 min and T = 8500 min, respectively, and we can conclude that the proposed model can accurately predict the trend change of future HI. The RUL prediction results demonstrated in Figure 10, the blue curve is a 20% confidence interval of the actual value, and we can see that the error between the predicted value and the actual value is small.

### 4.5. Comparison with Other Methods

In this part, three state-of-the-art approaches are implemented to predict HI and RUL of the bearing vibration signals for comparison, which includes BPNN, LSTM and GRU. The structure of these methods are chosen as 10-5-1, 64-10-1, and 64-10-1, respectively, and the learning rate is chosen as 0.001, and the number of iterations is chosen as 200. Figure 11 exhibits the HI prediction results of the four methods. It should be noted that the proposed method achieves the outstanding predictive performance, and the predicted HI is closer to the actual values than other approaches. In Figure 12, the curves of the proposed method, LSTM, and GRU mostly lie in the confidence interval, while the curves of BPNN overstepping the boundary in the later prediction stage. This indicates that the presented method outperforms the other three models, GRU slightly better than the LSTM, and BPNN has the maximum deviation. 

Afterwards, as tabulated in Table 2 and Table 3, the demonstrated results give further details. Table 2 shows the results of estimated RUL values at different times. It is clearly seen that the proposed method provides more accurate estimation results compared with other approaches. In Table 3, it can be summarized that the proposed prognostics model has shown an excellent prognostic performance compared with the other three approaches at different stages, according to the evaluation indicators of MAE, RMSE, R-Square and MAPE. 

## 5. Conclusions

The reliable prognosis technique for rolling bearing predictive maintenance and autonomic logistics is urgently needed in the industry. To address the important issues, we provided a reliable prognosis approach for degradation evaluation of rolling bearing in this study. The correlation kurtosis for different iteration periods and GPLVM is adopted to extract the special HI. Following this, a prognostics model, i.e., MCLSTM, is constructed and utilized for predicting future HI and RUL. The proposed MCLSTM is experimentally validated using the full-life-cycle vibration dataset of a rolling bearing. Through a comparison of different classical deep learning methods, the results validate that the proposed approach was more robust and effective than other approaches for HI and RUL estimation.

It should be noted that more extensive studies and complex operating conditions will validate the feasibility and effectiveness of the presented method in future work. In addition, the time to start prediction and threshold determination are both significant and challenging in HI and RUL prediction. 

## Figures and Tables

**Figure 1 sensors-20-01864-f001:**
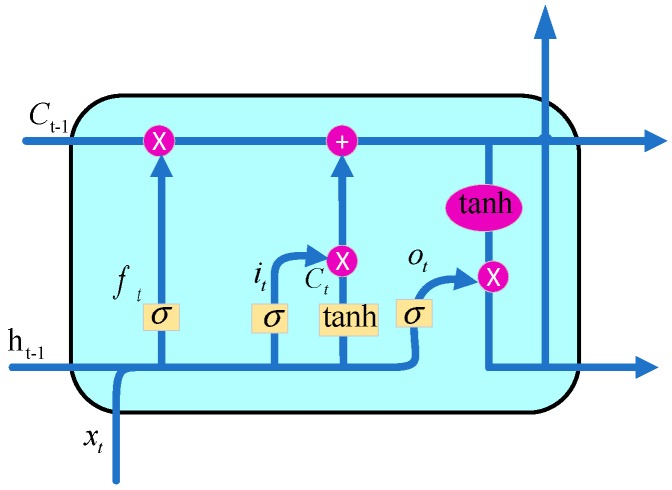
Long short-term memory (LSTM) hidden layer cell structure.

**Figure 2 sensors-20-01864-f002:**
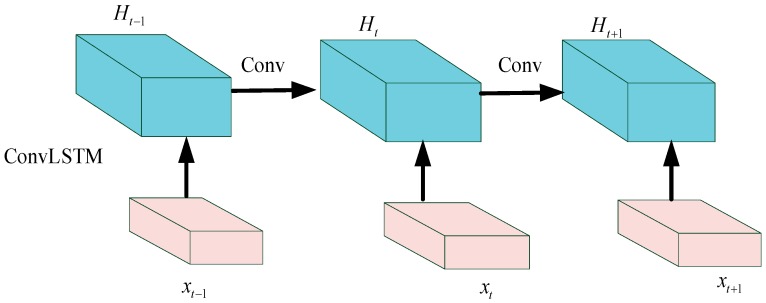
The ConvLSTM structure.

**Figure 3 sensors-20-01864-f003:**
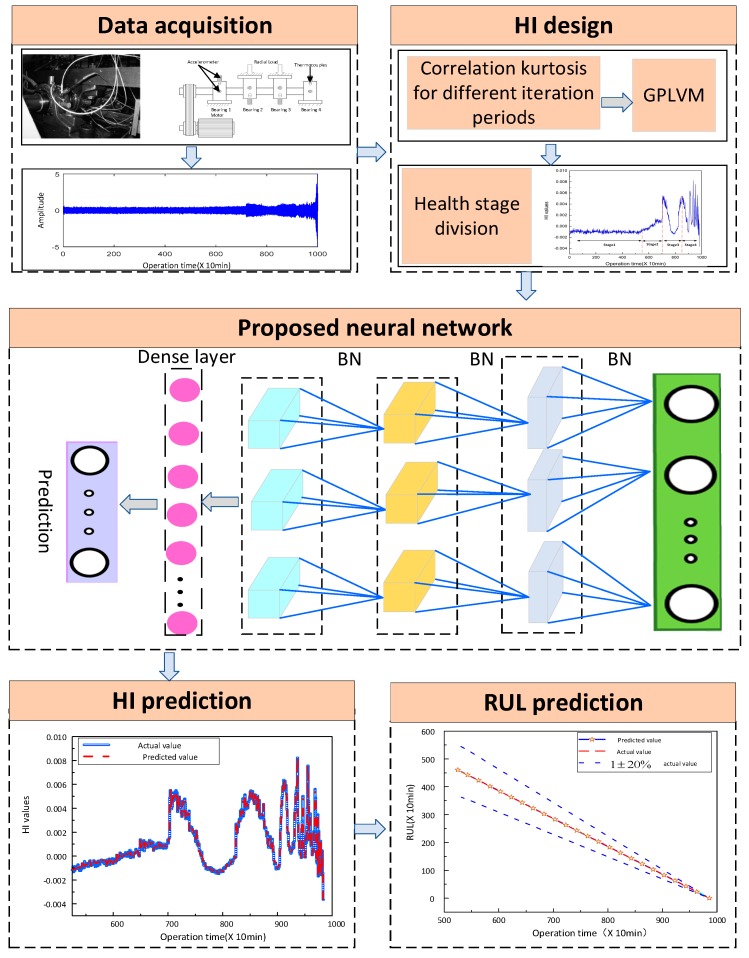
A flowchart of the proposed method.

**Figure 4 sensors-20-01864-f004:**
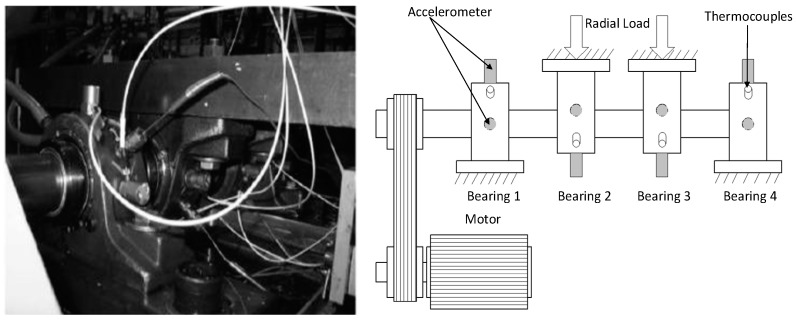
Test rig of Intelligent Maintenance Systems (IMS).

**Figure 5 sensors-20-01864-f005:**
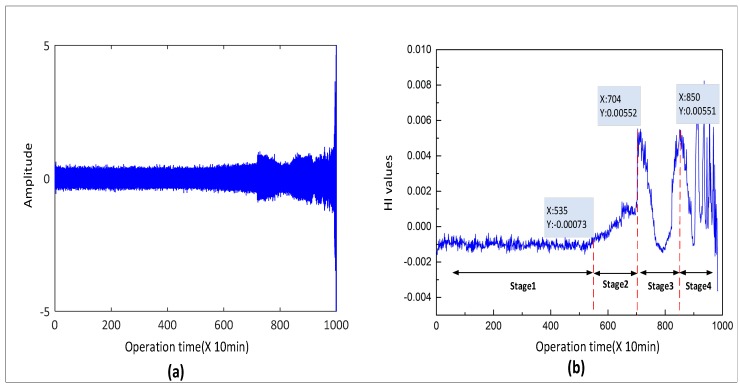
(**a**) Time-domain waveform of bearing, (**b**) health stage division of bearing.

**Figure 6 sensors-20-01864-f006:**
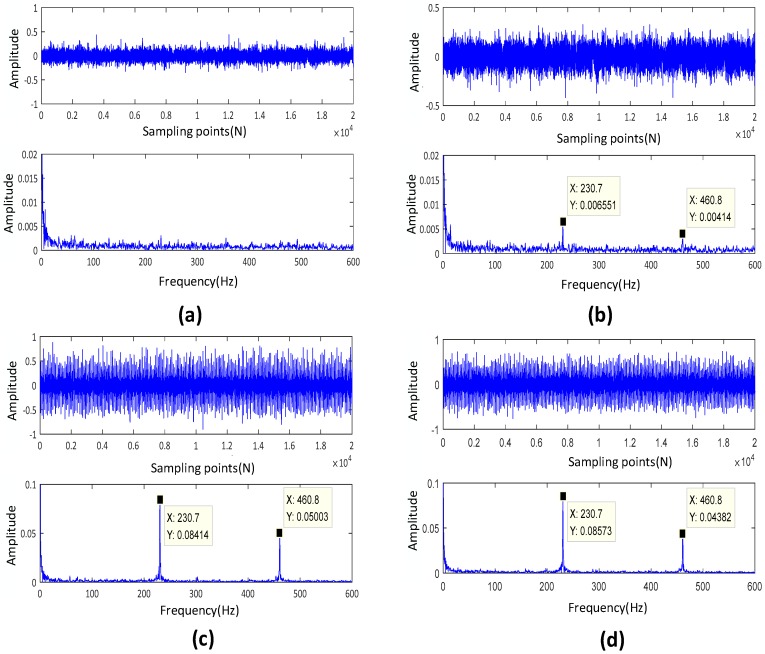
Time domain and envelope analysis of signals at different stages, (**a**) T = 550 min. (**b**) T = 5350 min, (**c**) T = 7040 min, (**b**) T = 8500min.

**Figure 7 sensors-20-01864-f007:**
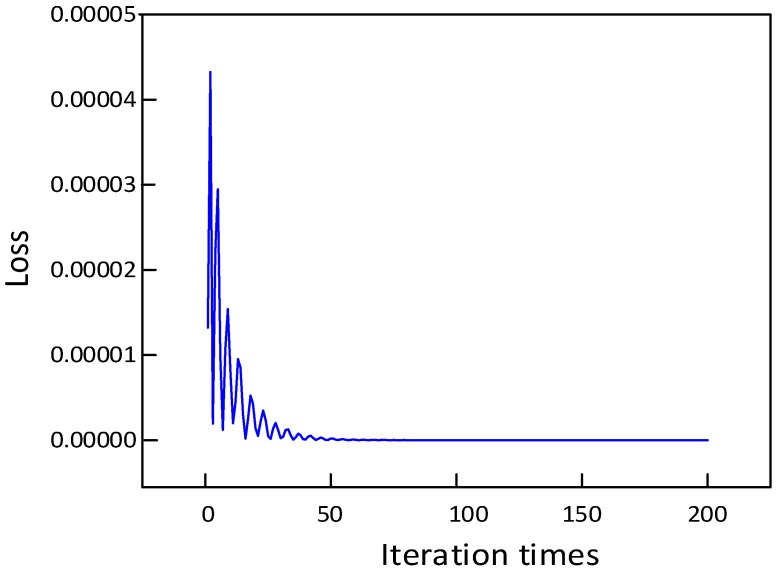
The loss-iteration times.

**Figure 8 sensors-20-01864-f008:**
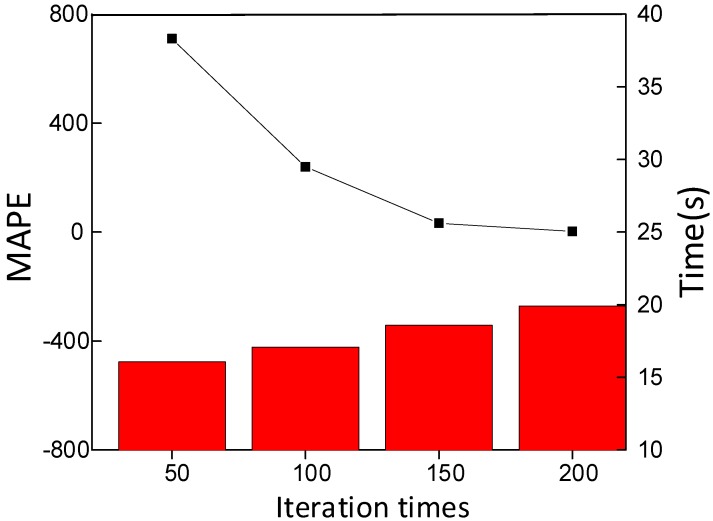
Influence of iterative times.

**Figure 9 sensors-20-01864-f009:**
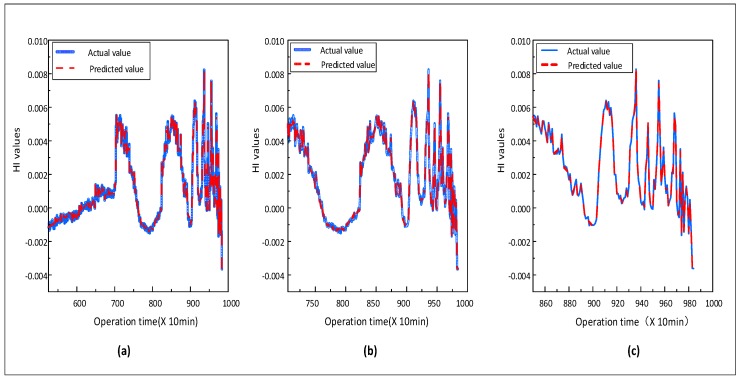
HI prediction start from different times, (**a**) T = 5350 min, (**b**) T = 7040 min, (**c**) T = 8500 min.

**Figure 10 sensors-20-01864-f010:**
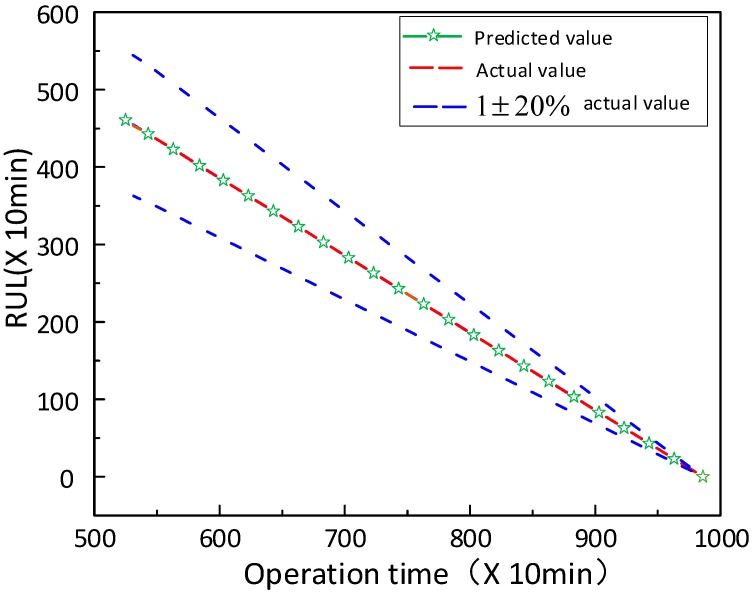
Remaining useful life (RUL) prediction of the proposed method.

**Figure 11 sensors-20-01864-f011:**
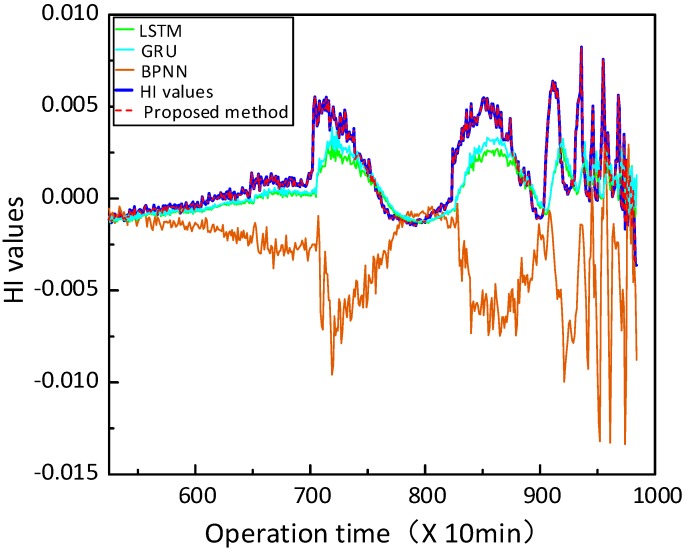
HI prediction results of different methods.

**Figure 12 sensors-20-01864-f012:**
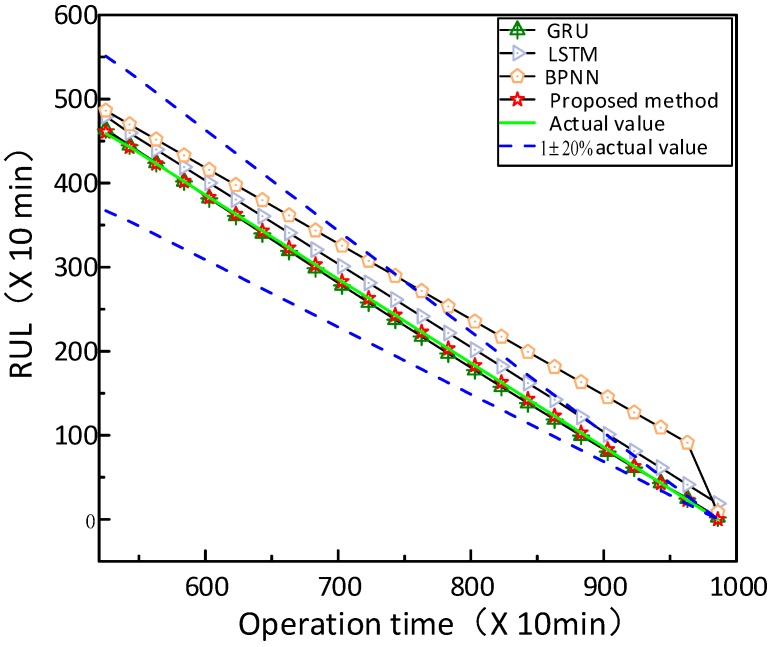
RUL prediction results of different methods.

**Table 1 sensors-20-01864-t001:** Evaluation indexes of different optimizers.

Optimizer	MAE	RMSE	R-Square	MAPE	Training Time(s)
Adam	$7.90\times 10^{-6}$	$7.91\times 10^{-3}$	0.999	1.257	19.07
RmsProp	$4.41\times 10^{-3}$	$4.41\times 10^{-3}$	−2.336	289.63	16.86
Adagrad	$6.47\times 10^{-3}$	$9.01\times 10^{-5}$	0.9187	118.4	16.88
SGD	$3.83\times 10^{-3}$	$4.38\times 10^{-3}$	−2.336	944.71	17.15

**Table 2 sensors-20-01864-t002:** Estimated RUL results of the four approaches.

Method	Inspection Time	Actual RUL	Estimated RUL	Prediction Error
GRU	540	444	448.2	−0.009
	704	280	283.1	−0.011
	982	2	3.788	−0.894
LSTM	540	444	462.6	−0.004
	704	280	289.3	−0.033
	982	2	4.117	−1.058
BPNN	540	444	472.5	−0.064
	704	280	329.1	−0.175
	982	2	8.335	−3.167
The proposed method	540	444	446.0	0.004
	704	280	280.8	−0.0028
	982	2	1.553	0.2235

**Table 3 sensors-20-01864-t003:** Performance comparison of four prognostic methods at different stage.

Method	Start Forecast time	MAE	RMSE	R-Square	MAPE
GRU	535	6.866	7.335	0.965	40.19
	705	14.89	17.10	0.954	58.51
	850	20.68	25.28	0.969	58.22
LSTM	535	13.88	17.68	0.982	44.76
	705	15.86	18.22	0.9487	62.79
	850	15.24	16.16	0.832	87.23
BPNN	535	47.54	49.27	0.862	89.78
	705	149.9	151.7	−2.550	389.6
	850	106.2	106.7	−6.285	471.1
The proposed method	535	0.9161	1.153	0.999	inf
	705	0.726	0.766	0.999	inf
	850	1.958	2.204	0.997	inf

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
