# Peer review of "A Reliable Prognosis Approach for Degradation Evaluation of Rolling Bearing Using MCLSTM"

_sensors, 2020, doi:10.3390/s20071864_

Round 1

Reviewer 1 Report

This paper presents a reliable prognosis approach for degradation evaluation of rolling bearing. By predicting the health indicator values and remaining useful life value of the rolling bearing, the results show the effectiveness of the proposed method.

The work is interesting. This method can be extended to the intelligent diagnosis of more complex mechanical equipment and has bright prospects in other machinery industry. However, the overall presentation and discussion could be improved as suggested below:

(1) The English level should be improved. The authors should check the paper in detail. For example, just to cite some and not all:

-  P.1 line 25: “tends to be” and not “tends to”

-  P.1 line 36: “based on” and not “based”

-  P.2 line 64: “predict” and not “predictive”

(2) In introduction part: please add a few more references on model-based approaches.

(3) The innovation of this article needs to be highlighted, please modify the introduction part.

(4)In figure 3,Please give the horizontal vertical axis unit in “HI prediction”, and the picture quality needs to be further improved.

(5) Why do the authors analyze the health stage division?

Author Response

Reviewer’s Comments:

This paper presents a reliable prognosis approach for degradation evaluation of rolling bearing. By predicting the health indicator values and remaining useful life value of the rolling bearing, the results show the effectiveness of the proposed method.

The work is interesting. This method can be extended to the intelligent diagnosis of more complex mechanical equipment and has bright prospects in other machinery industry. However, the overall presentation and discussion could be improved as suggested below:

1.The English level should be improved. The authors should check the paper in detail. For example, just to cite some and not all:

-  P.1 line 25: “tends to be” and not “tends to”

-  P.1 line 36: “based on” and not “based”

-  P.2 line 64: “predict” and not “predictive”

Authors’ Response:

Thanks the reviewer very much for the careful examination and correction. According to the reviewer’s advice, we carefully checked the language and expression of the manuscript and invited an English professor to work in reviewing team to improve the revised manuscript. The grammar errors and language problems throughout this manuscript were corrected earnestly, including the mentioned problems by the reviewer. All changes with a font color of RED are shown in the new revision.

2.In introduction part: please add a few more references on model-based approaches.

Authors’ Response:

It is really a valuable comment. Thanks the reviewer for this constructive comment.

In order to further improve the quality of the introduction, we have added three recent model-based articles based on the opinions of the reviewers. which are respectively:

[10]El-Tawil, K.; Jaoude, A.A.; Stochastic and nonlinear-based prognostic model, Syst. Sci. Control Eng. 2013, 1, 66–81.

[12] Wang, J.; Gao, R.X.; Yuan, Z.; Fan, Z.; Zhang, L.; A joint particle filter and expectation maximization approach to machine condition prognosis, J. Intell. Manuf. 2016,1–17.

[13] Qian, Y.; Yan, R.; Gao, R.X.; A multi-time scale approach to remaining useful life prediction in rolling bearing, Mech. Syst. Signal Process. 2017, 83, 549– 567.

All changes with a font color of RED are shown in the new revision.

3.The innovation of this article needs to be highlighted, please modify the introduction part.

Authors’ Response:

It is really a beneficial comment, and the authors thank the reviewer for this suggestion which have helped us improve this manuscript. With the development of sensor technology, the data of monitoring equipment is increasing. Traditional methods are difficult to predict the health of bearings from large amounts of data. With the development of sensor technology, data explosion has become a new problem of bearing remaining life prediction, which makes artificial intelligence technology develop rapidly in past years. In this paper, we proposed a reliable deep learning approach for degradation evaluation of rolling bearing. In order to effectively depict the degradation trend of rolling bearing, a special HI is constructed based on correlation kurtosis for different iteration periods and GPLVM. Then we present a novel deep learning network, i.e., MCLSTM, to predict the future HI and RUL. We verify the capability of the proposed approaches based on experimental dataset of rolling bearing. In order to further highlight the innovation of this article, we have added the contribution of this method to the revised manuscript. All changes with a font color of RED are shown in the new revision.

4.In figure 3,Please give the horizontal vertical axis unit in “HI prediction”, and the picture quality needs to be further improved.

Authors’ Response:

Thanks the reviewer very much for the careful examination. According to the reviewer’s advice, we carefully modify the picture quality. All changes are shown in the new revision.

5.Why do the authors analyze the health stage division?

Authors’ Response:

It is really a valuable question. The HI of rolling bearing generally present varying degradation trends with the development of fault severity. The degradation processes of machinery should be divided into different health stage according to the varying trends of HI before RUL prediction. Health stage division aims to divide the continuous degradation processes of machinery into different health stage according to the varying trends of HI [1]. Due to the variation of the degradation trends in different stages, multi-model prediction is expected to perform better than a single prognostic model [2]. Therefore, we think it is necessary to analyze the division of health stages.

The authors hope the above illustration can meet with the editor and reviewer’s kind approval.

Reference:

[1] H.-E. Kim, A.C.C. Tan, J. Mathew, B.-K. Choi, Bearing fault prognosis based on health state probability estimation, Expert Syst. Appl.2012, 39: 5200– 5213.

[2] Lei, Y.; Li, N.; Guo, L.; Li, N.; Yan, T.; Lin, J. Machinery health prognostics: a systematic review from data acquisition to RUL prediction. Mech. Syst. Signal Proc. 2018, 104, 799-834.

Reviewer 2 Report

This paper proposes a prognosis method for degradation evaluation of rolling bearing, which has practical applications. It is better that the authors improve the paper based on following suggestion:

  1. There are many spelling errors in the manuscript, such as, in page 2, Line 72, “essence” would be “essential”. Please check the manuscript carefully.
  2. Figure 5(a) and Figure 6 may have the problems,the symbol “Amplitude/m.s2” may be “Amplitude/m.s-2”. Please check it.
  3. The flow chart of proposed method is simple, please add more details and explanations.
  4. There many hyperparameter in deep learning model,why the authors choose the iteration times and optimizers?
  5. Please add some sentences about future analysis in conclusion.

Author Response

Reviewer’s Comments:

This paper proposes a prognosis method for degradation evaluation of rolling bearing, which has practical applications. It is better that the authors improve the paper based on following suggestion:

1.There are many spelling errors in the manuscript, such as, in page 2, Line 72, “essence” would be “essential”. Please check the manuscript carefully.

Authors’ Response:

Thanks for the reviewer’s careful examination and correction. We feel sorry for this mistake, in the revised manuscript all the typos and English errors scattered throughout the paper have been checked and modified carefully. All changes with a font color of RED are shown in the new revision.

2.Figure 5(a) and Figure 6 may have the problems,the symbol “Amplitude/m.s2” may be “Amplitude/m.s-2”. Please check it.

Authors’ Response:

Thanks for the reviewer’s careful examination. We feel sorry for this mistake, “Amplitude/m.s-2” is modify as “Amplitude” in Figure5(a) and Figure 6.

3.The flow chart of proposed method is simple, please add more details and explanations.

Authors’ Response:

Thanks for the reviewer’s constructive suggestion. According to the reviewer’s suggestion, we further analyze and explain the flow chart of the proposed method according to figure 3, including: data acquisition, HI design, Proposed neural network, HI and RUL prediction. In order to give a more clear description, the authors revised the “section 3” in the revision and all the changes are shown with a font color of RED.

4.There many hyperparameter in deep learning model,why the authors choose the iteration times and optimizers?

Authors’ Response:

It is really a valuable question. In fact, the author has carried out experiments on the proposed model by modifying the hyper parameters. We find that the number of iterations and optimizer have a great influence on the prediction results of the proposed model. Therefore, we analyze and discuss the influence of iteration times and optimizers selection on the prediction results of the model, and the experimental results also prove that the proposed model achieves an ideal prediction result. However, other hyper parameters may affect the accuracy of prediction, but the proposed method has achieved good results. In addition, limited by the length of the manuscript, the author didn’t analysis of other parameters in this manuscript.

The authors hope the above illustration can meet with the editor and reviewer’s kind approval.

5.Please add some sentences about future analysis in conclusion.

Authors’ Response:

Thanks for the reviewer’s constructive suggestion. According to the reviewer’s suggestion, we have added some content about the future work in the conclusion. All changes with a font color of RED are shown in the new revision.

Reviewer 3 Report

This article is very interesting; however, the authors need to take in consideration the following suggestions before to accept it:

  1. The explanation of Fig 3 needs to be increased or improved.
  2. The authors need to explain better the experimental setup.
  3. The authors need to justify the use of the four indexes to judge the performance of the different models for rolling remaining useful life prediction. Hence, I would like to know why these indexes? Why not others?
  4. The conclusion needs to be improved, adding quantitative results not only qualitative results. In addition, it is important to mention, what is the next with the investigation?

Author Response

Reviewer’s Comments:

This article is very interesting; however, the authors need to take in consideration the following suggestions before to accept it:

1.The explanation of Fig 3 needs to be increased or improved.

Authors’ Response:

Thanks for the reviewer’s careful examination and detailed suggestion. According to the reviewer’s advice, we carefully modify the picture quality. All changes are shown in the new revision.

2.The authors need to explain better the experimental setup.

Authors’ Response:

Thanks for the reviewer’s constructive suggestion. According to the reviewer’s advice, we have explained the experimental equipment in detail in the revised version with the font color of RED.

3.The authors need to justify the use of the four indexes to judge the performance of the different models for rolling remaining useful life prediction. Hence, I would like to know why these indexes? Why not others?

Authors’ Response:

It is really a valuable question. In deep learning prediction model, the MSE, RMSE, MAPE and R-squrae are common evaluation index to evaluate the prognostic performance of the approach [1]. MAE is more available for measuring the deviation between prediction values and true values. RMSE is sensitive to bigger values in a time series. MAPE is equivalent to normalizing the error of each point, reducing the impact of absolute error caused by mutation points. R-squre is an evaluation index of the data fitting results of the model. At the same time, these indexes were used to evaluate the remaining useful life estimation of rolling bearings [2]. Therefore, in order to evaluate the performance of the model more comprehensively, we chose these four indexes to evaluate the performance of the proposed model.

Refence:

[1] Sankalita S, Bhaskar S, Abhinav S, Kai G, Jose C. Metrics for offline evaluation of prognostic performance. Int. J. Progn. Health Manage 2010, 1(1), 1–20.

[2]Li X.; Zhang, W.;Ding, Q. Deep learning-based remaining useful life estimation of bearings using multi-scale feature extraction. Reliab. Eng. Syst. Safe 2019, 182, 208-218.

4.The conclusion needs to be improved, adding quantitative results not only qualitative results. In addition, it is important to mention, what is the next with the investigation?

Authors’ Response:

Thanks for the reviewer’s constructive suggestion. According to the reviewer’s advice, we have been added some quantitative results and some future work in our conclusion. The time to start prediction and threshold determination in HI and RUL prediction will be investigated in the future work. The conclusion has been carefully revised with the font color of RED and the authors hope the above changes can meet with the editor and reviewer’s kind approval.

Reviewer 4 Report

Dear Authors,

It was a pleasure to review your paper. Find below some observations regarding it:

1. The state-of-the-art presentation of the RUL determination of rolling bearings is comprehensive and based on relatively new references.

2. The theoretical background of the proposed method is well-written and easy to understand. The used figures are valuable.

3. I much appreciate the use of own experimental results instead of using data public available on the Internet.

4. A justification should be needed why acceleration signals were used for monitoring and not others, detailed in several papers, as

Henao, et al.: Trends in fault diagnosis for electrical machines – A review of diagnostic techniques;

Frosini, et al.: Induction machine bearing faults detection by means of statistical processing of the stray flux measurement;

Immovilli, et al.: Diagnosis of bearing faults in induction machines by vibration or current signals: a critical comparison.

5. Do not label axes with a ratio of quantities and units (as amplitude/ms-2).

6. In Fig. 8 the measurement unit of the time is missing. This is missing also from table I.

7. The conclusion should be not a simple overview of the paper or a reiteration of the given results. It should present the last word on the issues you raised in your paper, summarize your thoughts and conveying the larger implications of your study, demonstrate the importance of your ideas and introduce possible new or expanded ways of thinking about the research problem in discussion.

Author Response

Reviewer’s Comments:

It was a pleasure to review your paper. Find below some observations regarding it:

1.The state-of-the-art presentation of the RUL determination of rolling bearings is comprehensive and based on relatively new references.

Authors’ Response:

The authors thank the reviewer for the positive comments to our work.

2.The theoretical background of the proposed method is well-written and easy to understand. The used figures are valuable.

Authors’ Response:

The authors thank the reviewer for the positive comments to our work.

3.I much appreciate the use of own experimental results instead of using data public available on the Internet.

Authors’ Response:

The authors thank the reviewer for the positive comments to our work.

4.A justification should be needed why acceleration signals were used for monitoring and not others, detailed in several papers, as

Henao, et al.: Trends in fault diagnosis for electrical machines – A review of diagnostic techniques;

Frosini, et al.: Induction machine bearing faults detection by means of statistical processing of the stray flux measurement;

Immovilli, et al.: Diagnosis of bearing faults in induction machines by vibration or current signals: a critical comparison.

Authors’ Response:

It is really a valuable question. The vibration is often produced in the process of mechanical system operation, and the vibration signal contains rich fault feature information. In most practical engineering, the mechanical fault diagnosis based on vibration signal is one of the most commonly methods. The fault diagnosis technology based on vibration signal can real-timely, intuitively and accurately characterize the dynamic characteristics of equipment and the fault development process. But it also has its own drawbacks. When the bearing fails, the vibration signal contains a lot of noise, which will make it difficult to diagnose early wear and fatigue cracks of rolling bearing. However, this paper focuses on the prediction of the remaining useful life, rather than the fault diagnosis of rolling bearings. In addition, we can obtain the vibration signal more simply and quickly than other signals. Therefore, we are selected the acceleration signals to monitor the health condition of bearing.

The authors hope the above illustration can meet with the editor and reviewer’s kind approval.

5.Do not label axes with a ratio of quantities and units (as amplitude/ms-2).

Authors’ Response:

Thanks for the reviewer’s careful examination and detailed suggestion. The authors feel sorry for this mistake. The right label axes have been corrected in the revised manuscript. In addition, all the other label axes were checked again carefully by the authors.

6.In Fig. 8 the measurement unit of the time is missing. This is missing also from table I.

Authors’ Response:

Thanks for the reviewer’s careful examination and detailed suggestion. The measurement unit has been added in Figure 8 and table 1and shown with a font color of RED in the revised manuscript.

7.The conclusion should be not a simple overview of the paper or a reiteration of the given results. It should present the last word on the issues you raised in your paper, summarize your thoughts and conveying the larger implications of your study, demonstrate the importance of your ideas and introduce possible new or expanded ways of thinking about the research problem in discussion.

Authors’ Response:

Thanks for the reviewer’s constructive suggestion. According to the reviewer’s suggestion, we have has been carefully revised the conclusion. First, we have been present the issues to be solved in our paper, then summarize our thoughts and demonstrate the importance of our ideas. Finally, we discuss some possible new issue in the research in the future work. All changes with a font color of RED are shown in the new revision, and the authors hope the above changes can meet with the editor and reviewer’s kind approval.
